# ON SELF-SUPERVISED IMAGE REPRESENTATIONS FOR GAN EVALUATION

**Stanislav Morozov**
Marchuk Institute of Numerical Mathematics RAS
stanis-morozov@yandex.ru

**Andrey Voynov**
Yandex
avoin@yandex-team.ru

**Artem Babenko**
Yandex
HSE University
artem.babenko@phystech.edu

## ABSTRACT

The embeddings from CNNs pretrained on Imagenet classification are de-facto standard image representations for assessing GANs via FID, Precision and Recall measures. Despite broad previous criticism of their usage for non-Imagenet domains, these embeddings are still the top choice in most of the GAN literature.

In this paper, we advocate using state-of-the-art self-supervised representations to evaluate GANs on the established non-Imagenet benchmarks. These representations, typically obtained via contrastive or clustering-based approaches, provide better transfer to new tasks and domains, therefore, can serve as more universal embeddings of natural images. With extensive comparison of the recent GANs on the standard datasets, we demonstrate that self-supervised representations produce a more reasonable ranking of models in terms of FID/Precision/Recall, while the ranking with classification-pretrained embeddings often can be misleading. Furthermore, using self-supervised representations often improves the sample-efficiency of FID, which makes it more reliable in limited-data regimes.

## 1 INTRODUCTION

Generative adversarial networks (GANs) are an extremely active research direction in machine learning. The intensive development of the field requires established quantitative measures to assess constantly appearing models. While a large number of evaluation protocols were proposed (Borji, 2019; Xu et al., 2018; Zhou et al., 2019; Naeem et al., 2020), there is still no consensus regarding the best evaluation measure. Across the existing measures, the Fréchet Inception Distance (FID) (Heusel et al., 2017) and Precision/Recall (Kynkäänniemi et al., 2019) are the most widely adopted due to their simplicity and decent consistency with human judgments. FID and Precision/Recall quantify the discrepancy between distributions of real and generated images. Since these distributions are complicated to describe in the original RGB space, the images are represented by embeddings, typically extracted with CNNs pretrained on the Imagenet classification (Deng et al., 2009). While FID computed with these embeddings was shown to correlate with human evaluation (Heusel et al., 2017), these observations were mostly obtained on datasets, semantically close to Imagenet. Meanwhile, on non-Imagenet datasets, FID can result in inadequate evaluation, as widely reported in the literature (Rosca et al., 2017; Barratt & Sharma, 2018; Zhou et al., 2019).

In this work, we propose to employ the state-of-the-art self-supervised models (Chen et al., 2020a; He et al., 2020; Caron et al., 2020) to extract image embeddings for GAN evaluation. These models were shown to produce features that transfer better to new tasks, hence, they become a promising candidate to provide a more universal representation. Intuitively, classification-pretrained embeddings by design can suppress the information, irrelevant for the Imagenet class labels, which, however, can be crucial for other domains, like human faces. On the contrary, self-supervised models, mostly trained via contrastive or clustering-based learning, do not have such a bias since their main goal is typically to learn invariances to common image augmentations.

To justify the usage of self-supervised embeddings, we perform a thorough comparison of the recent GAN models trained on the five most common benchmark datasets. We demonstrate that classification-pretrained embeddings can lead to incorrect ranking in terms of FID, Precision, and Recall, which are the most popular metrics. On the other hand, self-supervised representations produce more sensible ranking, advocating their advantage over "classification-oriented" counterparts. Since all the checkpoints needed to compute self-supervised embeddings are publicly available, they can serve as a handy instrument for GAN comparison, consistent between different papers. We release the code for the "self-supervised" GAN evaluation along with data and human labeling reported in the paper online[1]. To sum up, the contributions of this paper are as follows:

1. To the best of our knowledge, our work is the first to employ self-supervised image representations to evaluate GANs trained on natural images.

2. By extensive experiments on the standard non-Imagenet benchmarks, we demonstrate that the usage of self-supervised representations provides a more reliable GAN comparison.

3. We show that the FID measure computed with self-supervised representations often has higher sample-efficiency and analyze the sources of this advantage.

## 2 RELATED WORK

**GAN evaluation measures.** Over the last years, a variety of quantitative GAN evaluation methods have been developed by the community, and the development process has yet to converge since all the measures possess specific weaknesses (Borji, 2019; Xu et al., 2018). The Inception Score (Salimans et al., 2016) was the first widely adopted measure but was shown to be hardly applicable for non-Imagenet domains (Barratt & Sharma, 2018). The Fréchet Inception Distance (FID) (Heusel et al., 2017) quantifies dissimilarity of real and generated distributions, computing the Wasserstein distance between their Gaussian approximations, and is currently the most popular scalar measure of GAN's quality. Several recent measures were proposed (Sajjadi et al., 2018; Kynkäänniemi et al., 2019; Naeem et al., 2020) that separately evaluate fidelity and diversity of GAN-produced images. All of them mostly use the embeddings produced by the Imagenet classification CNN. A recent work (Zhou et al., 2019) has introduced a human-in-the-loop measure, which is more reliable compared to automated ones but cannot be used, e.g., for monitoring the training process. We focus on three the most widely used measures: FID, Precision, and Recall, which are discussed briefly below.

**Fréchet Inception Distance** quantifies the discrepancy between the distributions of real and generated images, denoted by $p_D$ and $p_G$. Both $p_D$ and $p_G$ are defined on the high-dimensional image space forming nontrivial manifolds, which are challenging to approximate by simple functions. To be practical, FID operates in the lower-dimensional space of image embeddings. Formally, the embeddings are defined by a map $f : \mathbb{R}^N \to \mathbb{R}^d$, where $N$ and $d$ correspond to the dimensionalities of the images and embeddings spaces, respectively. By design, FID measures the dissimilarity between the induced distributions $fp_D, fp_G$ as follows. First, $fp_D$ and $fp_G$ are approximated by Gaussian distributions. Then the Wasserstein distance between these distributions is evaluated. As was shown in (Dowson & Landau, 1982), for distributions defined by the means $\mu_D, \mu_G$ and the covariance matrices $\Sigma_D, \Sigma_G$, this quantity equals to $\|\mu_D - \mu_G\|_2^2 + \mathrm{tr}(\Sigma_D + \Sigma_G - 2(\Sigma_D \Sigma_G)^{\frac{1}{2}})$. Lower FID values correspond to higher similarity between $p_G$ and $p_D$; hence, can be used to evaluate the performance of generative models. As a common practice in the FID computation, one typically uses the activations from the InceptionV3 (Szegedy et al., 2016) pretrained on Imagenet classification.

**Precision and Recall.** When assessing generative models, it is important to quantify both the visual quality of generated images and the model diversity, e.g., to diagnose mode collapsing. However, the scalar FID values were shown (Sajjadi et al., 2018; Kynkäänniemi et al., 2019) to sacrifice diversity in favor of visual quality, therefore FID cannot serve as the only sufficient metric. To this end, (Sajjadi et al., 2018) introduced Precision and Recall, which aim to measure the image realism and the model diversity, respectively. A recent follow-up (Kynkäänniemi et al., 2019) elaborates on these metrics and proposes a reasonable procedure to quantify both precision and recall based only on the image embeddings. In a nutshell, (Kynkäänniemi et al., 2019) assumes that the visual quality of a particular sample is high if its embedding is neighboring for the embeddings of the real images. On

---

[1]https://github.com/stanis-morozov/self-supervised-gan-eval

the other hand, a given real image is considered covered by the model if its embedding belongs to the neighborhood of embeddings of the generated images.

**Self-supervised representations.** Self-supervised learning is currently attracting much research attention, especially to contrastive learning and clustering-based methods (Chen et al., 2020a; He et al., 2020; Caron et al., 2020). The common idea behind these methods is to construct representations that are invariant to a wide range of common image augmentations. The recent self-supervised methods were shown to provide more transferrable (He et al., 2020; Caron et al., 2020) and robust (Hendrycks et al., 2019) features, which implies their usage as more universal representations. In this paper, we show them being a better alternative compared to established classifier-produced embeddings in the context of GAN assessment.

## 3 GAN EVALUATION

Here we systematically compare the publicly available GANs to highlight the cases of misleading comparison with classification-pretrained embeddings. Our goal is to demonstrate that self-supervised embeddings are a better alternative in these cases, while in other cases, the rankings with both types of embeddings are mostly consistent. We examine open-sourced GAN models[2] trained on five popular benchmarks:

- **CelebaHQ 1024x1024** (Karras et al., 2017) with the following GAN models: StyleGAN with truncation 0.7 (Karras et al., 2019a) and without it, MSG (Karnewar & Wang, 2020) with truncation 0.6 and without it, PGGAN (Karras et al., 2017). To compute the metrics, we use 30k real and synthetic images;
- **FFHQ 1024x1024** (Karras et al., 2019a) with the following GAN models: StyleGAN (Karras et al., 2019a), StyleGAN2 (Karras et al., 2019b), MSG (Karnewar & Wang, 2020) with truncation 0.6 and without it. To compute the metrics, we use 30k real and synthetic images;
- **LSUN Bedroom 256x256** (Yu et al., 2015) with the following GAN models: StyleGAN (Karras et al., 2019a) with truncation 0.7 and without it, PGGAN (Karras et al., 2017), COCO-GAN (Lin et al., 2019), RPGAN (Voynov & Babenko, 2019), RPGAN with high diversity (RPGAN div.). RPGAN generates 128x128 images, so we upscale them to 256x256. To compute the metrics, we use 30k real and synthetic images;
- **LSUN Church 256x256** (Yu et al., 2015) with the models: StyleGAN2 (Karras et al., 2019b) with truncation 0.5 and without it, MSG (Karnewar & Wang, 2020) with truncation 0.6 and without it, PGGAN (Karras et al., 2017), SNGAN (Miyato et al., 2018). SNGAN generates 128x128 images, so we upscale them to 256x256. To compute the metrics, we use 100k real and synthetic images;
- **Imagenet 128x128** (Deng et al., 2009) with the following GAN models: BigGAN (Brock et al., 2019), BigGAN-deep (Brock et al., 2019) (both with truncation 2.0), S3GAN (Lucic et al., 2019), Your Local GAN (YLG) (Daras et al., 2020). To compute the metrics, we use 50k images (50 per class). We include this dataset to demonstrate that for Imagenet, the proposed self-supervised representations provide consistent ranking with commonly used InceptionV3 embeddings.

To compute image embeddings, we use the following publicly available models:

- **InceptionV3** (Szegedy et al., 2016) pretrained on the ILSVRC-2012 task (Deng et al., 2009);
- **Resnet50** (He et al., 2016) pretrained on the ILSVRC-2012 task. We include this model since self-supervised models employ Resnet50, therefore, it is important to demonstrate that better GAN ranking comes from the training objective rather than the deeper architecture;
- **Imagenet21k** (Kolesnikov et al., 2019) pretrained on the multi-label classification task on approximately 14M images from the full Imagenet. Kolesnikov et al. (2019) have shown that supervised pretraining on huge datasets provides more transferrable features, therefore, **Imagenet21k** can also potentially provide more universal representations. The model architecture is Resnet50;
- **SwAV** (Caron et al., 2020) is the state-of-the-art self-supervised image representation model trained on ILSVRC-2012. The idea of SwAV is to simultaneously cluster the images while enforcing consistency between cluster assignments produced for different augmentations of the same image. The model architecture is Resnet50;

---

[2]The URLs for all models are provided in Appendix.

Table 1: FID values computed with different embeddings. The '*' symbol indicates models with truncation. The inconsistencies between InceptionV3 and SwAV rankings are highlighted in color.

**CELEBAHQ / FFHQ**

| | CELEBAHQ | | | | | FFHQ | | | |
|---|---|---|---|---|---|---|---|---|---|
| InceptionV3 | StyleGAN 5.958 | MSG 7.041 | PGGAN 7.747 | StyleGAN* 12.761 | MSG* 18.845 | StyleGAN2 3.355 | MSG 6.560 | StyleGAN 6.896 | MSG* 22.552 |
| Resnet50 | StyleGAN 5.981 | MSG 7.427 | PGGAN 9.395 | StyleGAN* 13.650 | MSG* 19.048 | StyleGAN2 3.813 | MSG 7.252 | StyleGAN 8.230 | MSG* 25.076 |
| Imagenet21k | MSG 295.3 | StyleGAN 301.3 | PGGAN 451.6 | StyleGAN* 635.5 | MSG* 874.4 | StyleGAN2 177.7 | MSG 334.7 | StyleGAN 389.1 | MSG* 1092 |
| SwAV | MSG 1.206 | StyleGAN 1.304 | StyleGAN* 1.473 | MSG* 1.832 | PGGAN 1.898 | StyleGAN2 0.634 | MSG 1.275 | StyleGAN 1.482 | MSG* 2.461 |
| DeepClusterV2 | MSG 1.847 | StyleGAN 2.255 | StyleGAN* 2.680 | PGGAN 2.865 | MSG* 2.935 | StyleGAN2 0.978 | MSG 1.890 | StyleGAN 2.076 | MSG* 3.926 |
| MoCoV2 | MSG 0.008 | StyleGAN 0.009 | PGGAN 0.012 | StyleGAN* 0.016 | MSG* 0.023 | StyleGAN2 0.005 | MSG 0.009 | StyleGAN 0.010 | MSG* 0.035 |

**LSUN-BEDROOM**

| | | | | | | |
|---|---|---|---|---|---|---|
| InceptionV3 | StyleGAN 2.986 | PGGAN 8.658 | StyleGAN* 9.655 | COCO-GAN 18.612 | RPGAN 37.924188 | RPGAN div. 40.165 |
| Resnet50 | StyleGAN 6.263 | PGGAN 17.689 | StyleGAN* 20.042 | COCO-GAN 34.045 | RPGAN 44.850 | RPGAN div. 51.850 |
| Imagenet21k | StyleGAN 321.7 | StyleGAN* 633.3 | PGGAN 940.6 | COCO-GAN 1270 | RPGAN div. 1589 | RPGAN 1593 |
| SwAV | StyleGAN 1.475 | StyleGAN* 1.776 | PGGAN 4.160 | RPGAN 5.608 | COCO-GAN 6.289 | RPGAN div. 6.460 |
| DeepClusterV2 | StyleGAN 2.095 | StyleGAN* 2.958 | PGGAN 5.855 | RPGAN 8.757 | COCO-GAN 9.151 | RPGAN div. 10.13 |
| MoCoV2 | StyleGAN 0.012 | StyleGAN* 0.031 | PGGAN 0.037 | RPGAN 0.072 | RPGAN div. 0.083 | COCO-GAN 0.085 |

**LSUN-CHURCH**

| | | | | | | |
|---|---|---|---|---|---|---|
| InceptionV3 | StyleGAN2 3.652 | MSG 5.009 | PGGAN 6.296 | MSG* 13.854 | StyleGAN2* 24.966 | SNGAN 32.661 |
| Resnet50 | StyleGAN2 6.650 | MSG 9.673 | PGGAN 10.850 | MSG* 22.670 | StyleGAN2* 55.571 | SNGAN 56.114 |
| Imagenet21k | StyleGAN2 558.1 | MSG 856.8 | PGGAN 946.0 | MSG* 1305.3 | SNGAN 1715 | StyleGAN2* 2165 |
| SwAV | StyleGAN2 1.898 | PGGAN 3.233 | MSG 3.578 | MSG* 4.062 | StyleGAN2* 5.194 | SNGAN 6.043 |
| DeepClusterV2 | StyleGAN2 2.599 | PGGAN 4.149 | MSG 4.586 | MSG* 5.395 | SNGAN 7.830 | StyleGAN2* 8.612 |
| MoCoV2 | StyleGAN2 0.019 | MSG 0.030 | PGGAN 0.030 | MSG* 0.048 | SNGAN 0.067 | StyleGAN2* 0.083 |

- **DeepClusterV2** (Caron et al., 2020) is another self-supervised model obtained by alternating between pseudo-labels generation via k-means clustering and training the network with a classification loss supervised by these pseudo-labels. The model architecture is Resnet50;

- **MoCoV2** (Chen et al., 2020b) is the state-of-the-art contrastive learning approach, which training objective enforces the closeness of representations produced for different augmentations of the same image while pushing apart the representations of unrelated images. The model architecture is Resnet50.

Three self-supervised models listed above outperform supervised pretraining on a number of transfer tasks (He et al., 2020; Caron et al., 2020), which implies that their embeddings capture more information relevant for these tasks, compared to supervised models pretrained on Imagenet. Below, for a large number of publicly available GANs, we present the values of FID, Precision, and Recall metrics computed with different embeddings. For the cases where the GANs ranking is inconsistent, we aim to show that the ranking obtained with the self-supervised representations is more reasonable.

Table 2: Prediction accuracy of CelebaHQ attributes from InceptionV3 and SwAV embeddings.

| Model | Mouth Slightly Open | No Beard | High Cheekbones | Smiling |
|---|---|---|---|---|
| InceptionV3 | 0.802 | 0.906 | 0.811 | 0.837 |
| SwAV | 0.868 | 0.938 | 0.851 | 0.893 |

## 3.1 FRÉCHET INCEPTION DISTANCE

The FID values for the non-Imagenet datasets computed with different embeddings are shown in Table 1. The cases of inconsistent ranking with supervised InceptionV3 and self-supervised SwAV embeddings are highlighted in color. The key observations are listed below:

**(a)** On CelebaHQ, SwAV ranks StyleGAN* higher, while InceptionV3/Resnet50 prefer PGGAN. Figure 1 shows random samples from both StyleGAN* and PGGAN and clearly demonstrates the superiority of StyleGAN*. To investigate the reasons why SwAV produces a more adequate ranking compared to Inception/Resnet50, we perform two additional experiments. *(I)* First, we verify that SwAV embeddings capture more information relevant for face images. The Celeba dataset (Liu et al., 2018) provides labels of 40 attributes for each image, describing various person properties (gender, age, hairstyle, etc.). For each attribute, we train 4-layer feedforward neural network with 2048 neurons on each layer with cross-entropy loss, which learns to predict the attribute from the SwAV/Inception embedding. For all attributes, the predictions from SwAV embeddings appear to be more accurate compared to InceptionV3 (several examples are given in Table 2). It confirms the intuition that InceptionV3 representations partially suppress the information about small facial details, which, however, is critical to identify more realistic images. *(II)* As a qualitative experiment, we compare SwAV and supervised Resnet50 embeddings visually via a recent technique described in Rombach et al. (2020). In a nutshell, this technique reveals the invariances learned by the particular representation model: for a given image, it visualizes several images having approximately the same embedding. By inspecting these images, one can analyze what factors of variations are not captured in the embedding (see the details in Section A.2). Two illustrative examples of such visualization for SwAV and Resnet50 are shown in Figure 2, demonstrating that Resnet50 embeddings are more invariant to sensitive information, like gender or race, compared to SwAV. Such ignorance of sensitive information makes supervised embeddings less appealing to use as universal representations. One of the key ingredients of the visualization method is an autoencoder, which is expected to capture all relevant information from an image. However, we argue that autoencoder representations are not well-suited for evaluating generative models and elaborate on this in detail in Section D.

**(b)** On Bedroom, there are two inconsistencies in InceptionV3 and SwAV ranking. The first is that SwAV ranks StyleGAN higher than PGGAN and the second is that SwAV ranks RPGAN higher than COCO-GAN. Figure 3 shows the samples from StyleGAN, PGGAN, RPGAN, and COCO-GAN and demonstrates that the ranking according to SwAV embeddings is more adequate. Namely, the quality of StyleGAN-generated images is substantially higher. Also, it is difficult to identify a favorite among RPGAN and COCO-GAN visually, while InceptionV3 embeddings claim strong superiority of the COCO-GAN model. On the other hand, self-supervised embeddings consider these models as comparable, which is better aligned with human perception.

**(c)** There are also cases of the inconsistent ranking of MSG and PGGAN on Church, and StyleGAN and MSG on CelebaHQ. But since the difference of the FID values are small for both InceptionV3 and SwAV, we do not consider it as a strong disagreement.

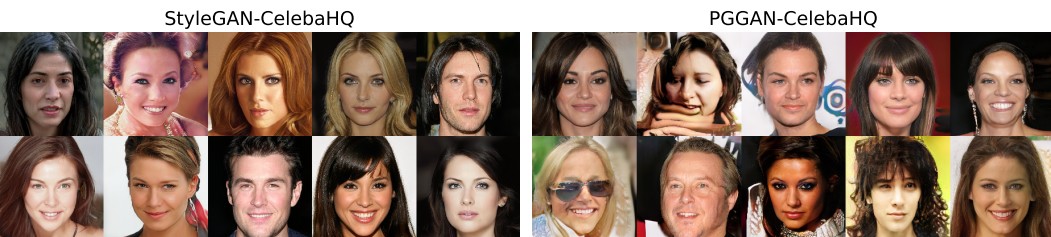

Figure 1: Samples generated by StyleGAN* and PGGAN trained on CelebaHQ. The quality of images generated by StyleGAN* is substantially higher.

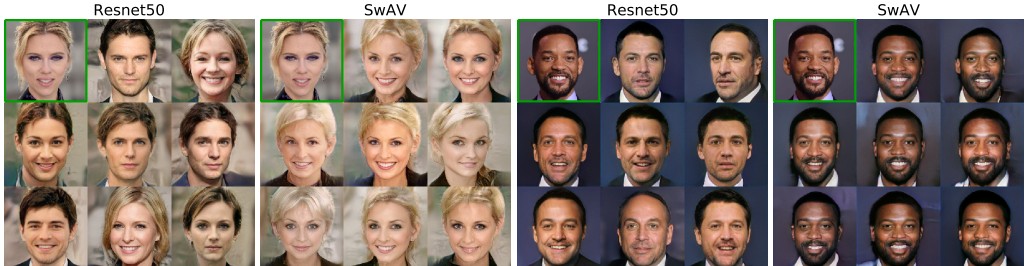

Figure 2: The reference image is top-left denoted by a green frame, while 8 others constitute a diverse sample of images that have approximately the same embeddings as the reference one.

**(d)** The rankings with the supervised InceptionV3/Resnet50 embeddings are consistent with each other while being different from the rankings with the self-supervised ones. It indicates that the model architecture does not affect GAN ranking, and it is the training objective that matters.

**(e)** Imagenet21k corrects some cases of misleading ranking with InceptionV3, but not all of them. Namely, it correctly ranks StyleGAN and PGGAN on Bedroom while being wrong on CelebaHQ.

**(f)** SwAV and DeepClusterV2 have minimal inconsistencies in the ranking of MSG* vs PGGAN on CelebaHQ and StyleGAN2* vs SNGAN on Church, but the differences in the absolute values of the FID metric are negligible, so we consider these embedding models as mostly consistent.

**(g)** MoCoV2 fixes some ranking mistakes with InceptionV3, but not all of them. While it fixes the ranking of StyleGAN and PGGAN on Bedroom and reduces the gap between RPGAN and COCO-GAN, the ranking of PGGAN and StyleGAN on CelebaHQ is still incorrect. Overall, the most reasonable rankings are obtained using SwAV/DeepCluster, which have significantly higher transfer performance compared to MoCoV2. In further experiments, we focus on the most transferable SwAV/DeepCluster models.

Overall, self-supervised embeddings provide a more reasonable FID ranking across existing non-Imagenet benchmarks. For completeness, we also report the FID values for the Imagenet dataset in Table 6. In this case, rankings with all embeddings are the same, which confirms that the SwAV representations can be used for Imagenet as well, while it is not the main focus of our work.

## 3.2 PRECISION

The values of the Precision metric are reported in Table 3. The main observations are listed below:

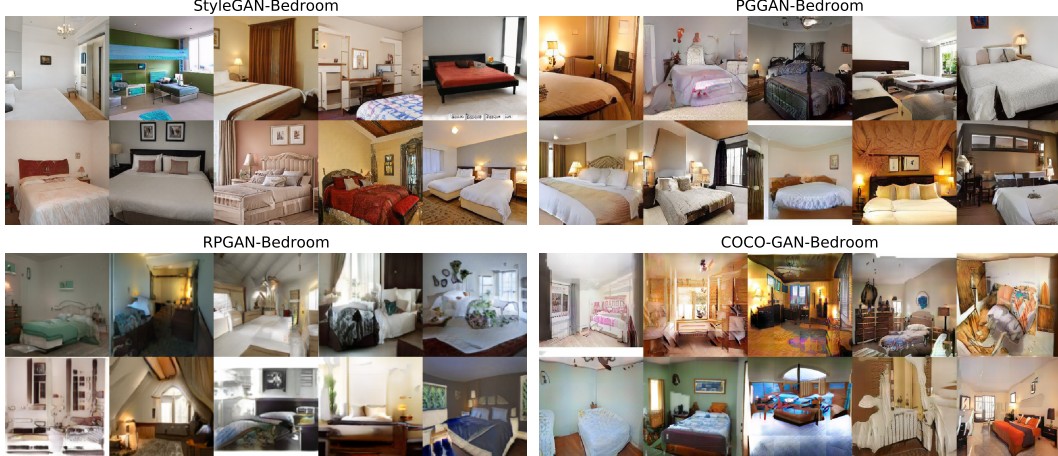

Figure 3: Samples generated by StyleGAN*, PGGAN, RPGAN and COCO-GAN trained on Bedroom. The quality of images generated by StyleGAN* is substantially higher, while the quality of the images generated by RPGAN and COCO-GAN is approximately the same.

Table 3: Precision ($k=5$) for different embedding models. The '*' symbol indicates models with truncation. Inconsistencies between InceptionV3 and SwAV models are highlighted in color.

| | CELEBAHQ | | | | | FFHQ | | | |
|---|---|---|---|---|---|---|---|---|---|
| InceptionV3 | MSG* 0.894 | StyleGAN* 0.882 | StyleGAN 0.803 | MSG 0.7963 | PGGAN 0.778 | MSG* 0.843 | StyleGAN2 0.778 | StyleGAN 0.778 | MSG 0.776 |
| Resnet50 | StyleGAN* 0.921 | MSG* 0.916 | StyleGAN 0.857 | MSG 0.839 | PGGAN 0.839 | MSG* 0.865 | StyleGAN2 0.821 | StyleGAN 0.807 | MSG 0.789 |
| Imagenet21k | StyleGAN* 0.923 | MSG* 0.912 | PGGAN 0.904 | StyleGAN 0.896 | MSG 0.868 | StyleGAN2 0.883 | MSG* 0.875 | StyleGAN 0.871 | MSG 0.851 |
| SwAV | MSG* 0.935 | StyleGAN* 0.928 | MSG 0.879 | StyleGAN 0.875 | PGGAN 0.851 | MSG* 0.911 | StyleGAN 0.905 | StyleGAN2 0.889 | MSG 0.870 |
| DeepClusterV2 | MSG* 0.945 | StyleGAN* 0.937 | StyleGAN 0.892 | PGGAN 0.890 | MSG 0.887 | MSG* 0.920 | StyleGAN 0.908 | StyleGAN2 0.882 | MSG 0.873 |

| | LSUN-BEDROOM | | | | | |
|---|---|---|---|---|---|---|
| InceptionV3 | StyleGAN* 0.799 | StyleGAN 0.649 | PGGAN 0.541 | COCO-GAN 0.443 | RPGAN div. 0.092 | RPGAN 0.086 |
| Resnet50 | StyleGAN* 0.849 | StyleGAN 0.722 | PGGAN 0.646 | COCO-GAN 0.596 | RPGAN div. 0.545 | RPGAN 0.515 |
| Imagenet21k | StyleGAN* 0.723 | StyleGAN 0.653 | COCO-GAN 0.622 | PGGAN 0.434 | RPGAN div. 0.171 | RPGAN 0.114 |
| SwAV | StyleGAN* 0.849 | StyleGAN 0.760 | PGGAN 0.732 | COCO-GAN 0.725 | RPGAN div. 0.540 | RPGAN 0.440 |
| DeepClusterV2 | StyleGAN* 0.817 | StyleGAN 0.728 | PGGAN 0.722 | COCO-GAN 0.706 | RPGAN div. 0.540 | RPGAN 0.423 |

| | LSUN-CHURCH | | | | | |
|---|---|---|---|---|---|---|
| InceptionV3 | StyleGAN2* 0.909 | MSG* 0.837 | MSG 0.698 | PGGAN 0.692 | StyleGAN2 0.689 | SNGAN 0.193 |
| Resnet50 | StyleGAN2* 0.910 | MSG* 0.828 | MSG 0.679 | StyleGAN2 0.672 | PGGAN 0.639 | SNGAN 0.403 |
| Imagenet21k | StyleGAN2* 0.567 | StyleGAN2 0.492 | MSG* 0.458 | MSG 0.448 | PGGAN 0.399 | SNGAN 0.183 |
| SwAV | StyleGAN2* 0.929 | MSG* 0.647 | StyleGAN2 0.619 | PGGAN 0.533 | MSG 0.491 | SNGAN 0.465 |
| DeepClusterV2 | StyleGAN2* 0.936 | MSG* 0.779 | StyleGAN2 0.630 | MSG 0.593 | PGGAN 0.589 | SNGAN 0.438 |

**(a)** As with FID, all supervised InceptionV3/Resnet50 embeddings provide the same ranking, except minor differences between MSG with truncation and StyleGAN on CelebaHQ, and StyleGAN2 and PGGAN on Church. Self-supervised SwAV and DeepClusterV2 are also consistent except for the negligible difference in the ranking of PGGAN and MSG on CelebaHQ and Church;

**(b)** The most notable inconsistency between supervised and self-supervised embeddings is revealed on LSUN-Church, where InceptionV3 considers MSG to be comparable to StyleGAN2, while SwAV ranks StyleGAN2 significantly higher. *(I)* To analyze which ranking of two GANs is more reasonable, we perform the following. On the synthetic data from the first GAN, we train a classifier that aims to distinguish between real and synthetic images. This classifier is then evaluated on the synthetic data from the second GAN. Concretely, we train a classifier to detect synthetic images on real LSUN-Church and the images generated by MSG. Then we evaluate this model on hold-out real images and images produced by StyleGAN2. Intuitively, if a model was trained on high-quality synthetic samples, it will easily detect lower-quality ones. On the other hand, if the model learns to detect only low-quality synthetics, it will be harder to discriminate real images from the high-quality ones. In this experiment, we employ a Resnet50 classifier with a binary cross-entropy loss. The results for Church are provided in Table 4, meaning that the StyleGAN2 images are of higher quality, therefore, the SwAV ranking is more reasonable. *(II)* We also conduct a human study to determine which of the generative models gives more realistic images in terms of human perception. For each generative model, we show ten assessors a real or randomly generated (fake) image and ask them to choose whether it is real or fake. The error rate reflects the visual quality of the generative model. For both models, MSG and StyleGAN2, we demonstrate to assessors 500 images, and the error rate is $0.4\%$ for MSG and $2.8\%$ for StyleGAN2, which clearly shows the superiority of StyleGAN2.

Table 5: Recall ($k$=5) for different GAN and embedding models. The '*' symbol indicates models with truncation. Inconsistencies between InceptionV3 and SwAV models are highlighted in color.

**CELEBAHQ**

| | | | | | | **FFHQ** | | | |
|---|---|---|---|---|---|---|---|---|---|
| InceptionV3 | MSG 0.477 | StyleGAN 0.405 | PGGAN 0.398 | StyleGAN* 0.281 | MSG* 0.276 | StyleGAN2 0.632 | MSG 0.538 | StyleGAN 0.474 | MSG* 0.339 |
| Resnet50 | MSG 0.502 | StyleGAN 0.419 | PGGAN 0.330 | StyleGAN* 0.284 | MSG* 0.277 | StyleGAN2 0.624 | MSG 0.547 | StyleGAN 0.456 | MSG* 0.299 |
| Imagenet21k | MSG 0.546 | StyleGAN 0.414 | MSG* 0.321 | StyleGAN* 0.315 | PGGAN 0.287 | StyleGAN2 0.688 | MSG 0.551 | StyleGAN 0.452 | MSG* 0.335 |
| SwAV | MSG 0.151 | StyleGAN 0.061 | MSG* 0.055 | StyleGAN* 0.035 | PGGAN 0.027 | StyleGAN2 0.399 | MSG 0.193 | StyleGAN 0.095 | MSG* 0.063 |
| DeepClusterV2 | MSG 0.185 | MSG* 0.074 | StyleGAN 0.072 | StyleGAN* 0.042 | PGGAN 0.039 | StyleGAN2 0.409 | MSG 0.220 | StyleGAN 0.132 | MSG* 0.078 |

**LSUN-BEDROOM**

| | | | | | | |
|---|---|---|---|---|---|---|
| InceptionV3 | StyleGAN 0.592 | PGGAN 0.516 | COCO-GAN 0.476 | StyleGAN* 0.397 | RPGAN 0.190 | RPGAN div. 0.110 |
| Resnet50 | StyleGAN 0.519 | PGGAN 0.352 | StyleGAN* 0.304 | COCO-GAN 0.258 | RPGAN 0.058 | RPGAN div. 0.030 |
| Imagenet21k | StyleGAN 0.498 | StyleGAN* 0.293 | PGGAN 0.240 | COCO-GAN 0.157 | RPGAN 0.008 | RPGAN div. 0.003 |
| SwAV | StyleGAN 0.153 | StyleGAN* 0.069 | PGGAN 0.015 | COCO-GAN 0.003 | RPGAN 0.0003 | RPGAN div. 0.00003 |
| DeepClusterV2 | StyleGAN 0.210 | StyleGAN* 0.109 | PGGAN 0.039 | COCO-GAN 0.009 | RPGAN 0.001 | RPGAN div. 0.0 |

**LSUN-CHURCH**

| | | | | | | |
|---|---|---|---|---|---|---|
| InceptionV3 | StyleGAN2 0.519 | MSG 0.491 | PGGAN 0.461 | MSG* 0.271 | SNGAN 0.090 | StyleGAN2* 0.071 |
| Resnet50 | StyleGAN2 0.460 | PGGAN 0.413 | MSG 0.406 | MSG* 0.182 | SNGAN 0.038 | StyleGAN2* 0.021 |
| Imagenet21k | StyleGAN2 0.350 | MSG 0.260 | PGGAN 0.251 | MSG* 0.105 | StyleGAN2* 0.013 | SNGAN 0.010 |
| SwAV | StyleGAN2 0.073 | PGGAN 0.019 | MSG 0.008 | MSG* 0.002 | StyleGAN2* 0.0014 | SNGAN 0.0002 |
| DeepClusterV2 | StyleGAN2 0.112 | PGGAN 0.058 | MSG 0.031 | MSG* 0.009 | SNGAN 0.002 | StyleGAN2* 0.002 |

**(c)** Imagenet21k ranks GANs less reliably compared to SwAV. The most notable mistake is a ranking of COCO-GAN and PGGAN on Bedroom, where PGGAN produces more visually appealing images, see Figure 3. Another case is comparison on CelebaHQ, where Imagenet21k ranks PGGAN higher than the more powerful MSG, see samples in Section B.

Table 4: The accuracy of fake images detection on Church. The rows correspond to GANs producing the train synthetics, while the columns correspond to GANs producing the test.

| Train/Test | MSG | StyleGAN2 |
|---|---|---|
| **MSG** | 0.999 | 0.610 |
| **StyleGAN2** | 0.967 | 0.979 |

## 3.3 RECALL

The values of the Recall metric are shown in Table 5, and the main observations are provided below:

**(a)** As in previous experiments, there are only minor inconsistencies between supervised InceptionV3 and Resnet50 models, namely, StyleGAN vs COCO-GAN on Bedroom and MSG vs PGGAN on Church. The only insignificant difference between the self-supervised methods is the ranking of StyleGAN with truncation vs SNGAN on Church, however, Recall values for both models are negligible. In terms of Recall, Imagenet21k ranking always coincides with the ranking obtained by self-supervised methods, except for the negligible discrepancy between MSG and PGGAN on Church;

**(b)** The absolute Recall values for SwAV/DeepClusterV2 are smaller compared to InceptionV3/Resnet50. We attribute this behavior to the fact that GANs tend to simplify images omitting the details (Bau et al., 2019), e.g., people in front of buildings, cars, fences, etc. The classifier-pretrained embeddings are less sensitive to these details since they are not crucial for correct classification. In contrast, self-supervised embeddings are more susceptible to small details (see Figure 2 and Table 4), hence, more images are considered "not covered". Figure 6 in Section C shows examples of real LSUN-Church images that are "definitely covered" by StyleGAN2 from the standpoint of InceptionV3 embeddings, but are "not covered" if SwAV embeddings are used. More formally, we say that a real image is covered by a synthetic one with the neighborhood size $k$, if the distance between their embeddings does not exceed the distance from the embedding of the synthetic image to its $k$-th nearest neighbor in the set of all synthetic embeddings. The images from Figure 6 are covered by at least 10 synthetic images with neighborhood size 5 with InceptionV3 embeddings, while being not covered even by the neighborhood of size 100 for SwAV embeddings. These images possess many small details, such as monuments, cars, people, branches in the foreground, and so on, that GANs usually omit to generate.

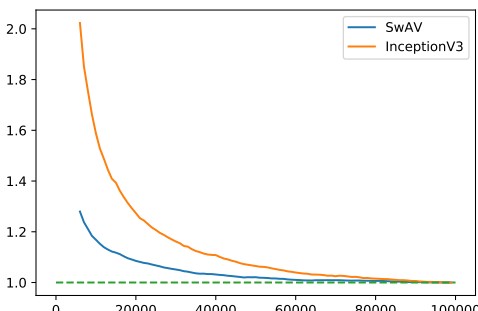

**(c)** Interestingly, the Precision values for SwAV are quite high, therefore, SwAV considers the state-of-the-art generative models being able to produce high-fidelity images, but failing to generate diverse images with small details.

**(d)** There are two significant inconsistencies in rankings based on InceptionV3 and SwAV. First, on CelebaHQ, SwAV prefers MSG* over PGGAN. Second, on LSUN-Bedroom, InceptionV3 ranks PG-GAN higher than StyleGAN. Since StyleGAN is a more powerful model compared to PGGAN, we also consider it as a case of a more reliable ranking with SwAV, even though it is difficult to confirm quantitatively due to the lack of an "oracle" measure of generation diversity.

Figure 4: FID values for different sample sizes for StyleGAN2 on Church. Since FID values for SwAV and InceptionV3 have different scales, they normalized by the FID value computed for a sample of size 100k.

### 3.4 SAMPLE EFFICIENCY OF SwAV-BASED FID

As an additional practical advantage of SwAV, we highlight that computing FID becomes more sample-efficient compared to InceptionV3. Namely, to obtain a reliable estimation of FID values, one requires much fewer samples when using SwAV embeddings. We illustrate this effect in Figure 4, which plots FID values w.r.t. sample size for StyleGAN2 trained on Church. Since the FID values for SwAV and InceptionV3 have different typical scales, we normalize both curves by the corresponding FID value computed for a sample of size 100k. FID based on SwAV embeddings converges faster, i.e., using SwAV always achieves more reliable FID estimates for a fixed sample size.

We attribute this benefit of SwAV to the fact that its representations capture more information needed to distinguish between real and fake distributions. Intuitively, the covariance matrices for real and synthetic data computed from SwAV embeddings are more dissimilar compared to InceptionV3-based ones. Quantitatively, the magnitude of the covariance term in FID $\mathrm{tr}(C_R + C_S - 2\sqrt{C_R C_S})$ is larger for SwAV, which leads to smaller relative errors of its stochastic estimates. We elaborate on this issue more rigorously in Section E.

## 4 CONCLUSION

In this paper, we have investigated if the state-of-the-art self-supervised models can produce more appropriate representations for GAN evaluation. With extensive experiments, we have shown that using these representations often corrects the cases of misleading ranking obtained with classification-pretrained embeddings. Overall, self-supervised representations provide a more adequate GAN comparison on the four established non-Imagenet benchmarks of natural images. Of course, we do not claim that they should be used universally for all areas, e.g., for spectrograms or medical images. But our work indicates that obtaining good representations needed for proper GAN evaluation does not require supervision, therefore, domain-specific self-supervised learning becomes a promising direction for further study.

## 5    ACKNOWLEDGEMENTS

We thank the Anonymous Reviewers for their reviews. We also thank Xun Huang for commenting on his experience with SwAV on OpenReview.

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

## A APPENDIX

### A.1 IMAGENET

Table 6: FID values for GANs on Imagenet.

| | IMAGENET | | | |
|---|---|---|---|---|
| InceptionV3 | BigGAN-deep | BigGAN | S3GAN | YLG |
| | 6.923 | 7.659 | 10.85 | 20.64 |
| Resnet50 | BigGAN-deep | BigGAN | S3GAN | YLG |
| | 11.22 | 19.50 | 20.05 | 34.11 |
| Imagenet21k | BigGAN-deep | BigGAN | S3GAN | YLG |
| | 334.8 | 601.5 | 1202 | 1697 |
| SwAV | BigGAN-deep | BigGAN | S3GAN | YLG |
| | 2.374 | 4.162 | 4.282 | 5.965 |
| DeepClusterV2 | BigGAN-deep | BigGAN | S3GAN | YLG |
| | 3.937 | 6.240 | 6.555 | 9.101 |

### A.2 THE DETAILS OF THE VISUALIZATION TECHNIQUE IN SECTION 3.1

The technique from Rombach et al. (2020) proposes to construct the embedding visualization in the following way. Let us have an autoencoder $A(x)$, whose latent representations are denoted by $z$. It is assumed that a latent representation $z=A(x)$ captures all the information from the image $x$, since the autoencoder's goal is to fully reconstruct $x$ from $z$. Let us denote the embedding model we want to study by $E(x)$. For a particular image $x$, Rombach et al. (2020) compute an autoencoder representation $z = A(x)$ and aims to disentangle the information contained in the autoencoder representation $z$ into the information contained in the image embedding $e = E(x)$ and the information $v$ to which the embedding model $E(x)$ is invariant. To this end, an invertible neural network (INN) is trained, which predicts the invariant part $v = \text{INN}(z, e)$ based on the autoencoder representation and embedding. Moreover, it is trained in such a way that the invariant part has a normal distribution $v \sim \mathcal{N}(0, 1)$. Finally, a bunch of images having the same $e$ but different invariant parts $v$ are produced. Namely, several samples of invariant parts $\hat{v}$ are produced from $\mathcal{N}(0, 1)$ and the corresponding latent representations are computed from $\hat{v}$ and embedding $e$, that is $\hat{z} = \text{INN.reverse}(v, e)$. Then $\hat{z}$ can be decoded into an image using the autoencoder. As a result, for a given embedding model $E(x)$ and a given image $x$, it allows to obtain a set of images with approximately the same embeddings but varying invariant part. For the CelebaHQ dataset, we use ALAE autoencoder (Pidhorskyi et al., 2020) and train INN with the hyperparameters provided in (Rombach et al., 2020).

## B CELEBAHQ SAMPLES

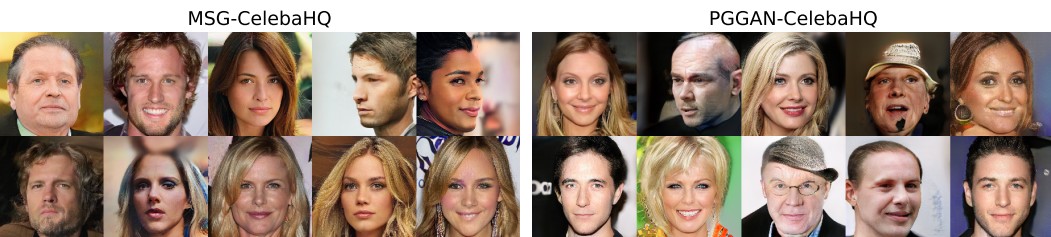

Figure 5: Random sample of images generated by MSG without truncation and PGGAN trained on CelebaHQ dataset. One can see that the quality of images generated by MSG is substantially higher.

## C  CHURCH IMAGES

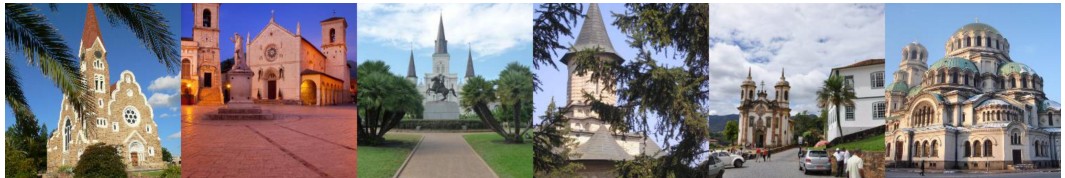

Figure 6: Examples of real images that are confidently covered by StyleGANv2 in terms of Inception V3 embeddings, but not covered in terms of SwAV.

## D  AUTOENCODER EMBEDDINGS

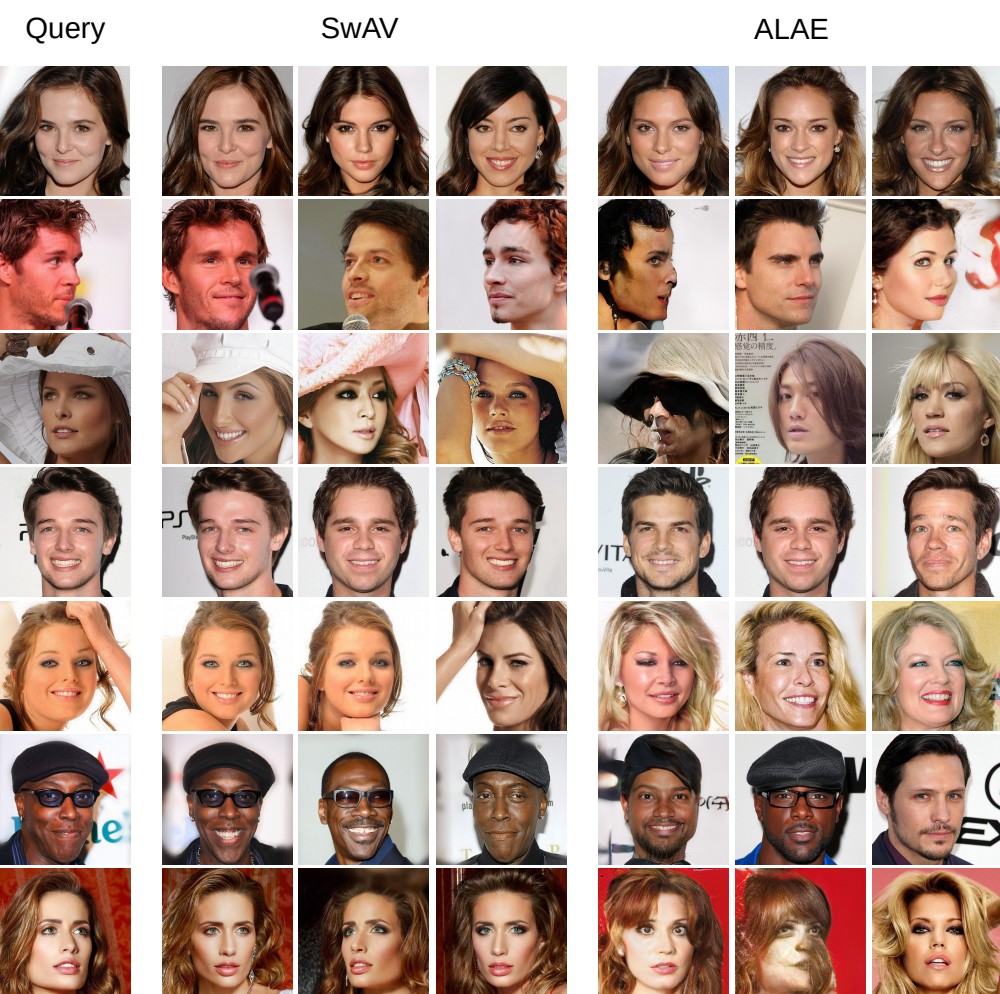

Figure 7: Examples of nearest neighbors in terms of SwAV and ALAE representations.

In this section, we compare SwAV and ALAE autoencoder (Pidhorskyi et al., 2020) embeddings. ALAE was trained on Celeba (Liu et al., 2018) dataset, therefore, is expected to work correctly for the ranking of generative models on CelebaHQ. To investigate what information is important for each of the embeddings, we build a dataset containing 30k real images from CelebaHQ and 30k synthetic images generated by PGGAN. Then we select 1.5k real and 1.5k synthetic images as queries and leave the remaining 57k pictures as a database. For each query, we compute three nearest neighbors from the database. Typical examples of the nearest neighbors are shown on Figure 7.

One can see that ALAE places a strong emphasis on the exact spatial arrangement, while sparsely sampled manifolds rarely include near-exact matches in terms of spatial structure (Kynkäänniemi et al., 2019). This makes ALAE embeddings poor for GAN evaluation, for instance, via Precision and Recall metrics. We also perform real/fake classification of queries using 1-NN classification on the constructed database. The classification accuracy is 0.729 for ALAE embeddings and 0.839 for SwAV. Overall, the distance in the space of the autoencoder's embeddings is less informative to distinguish between the real/fake distributions.

## E    ON THE ESTIMATE OF FID VALUES

Let us denote $C_R$ the covariance matrix for real data, $C_S$ for synthetic data and $\widetilde{C}_R$ and $\widetilde{C}_S$ their estimates with a finite number of samples. We also denote

$$d(C_R, C_S) = \text{tr}(C_R + C_S - 2\sqrt{C_R C_S}) \tag{1}$$

In (Dowson & Landau, 1982) it was shown that $d(C_R, C_S)$ defines a metric on the space of all covariance matrices of order $n$. Our goal is to assess the relative error of the FID estimate. Under the assumption that the means of real and synthetic data distributions are estimated quite accurately (which is the case already for a few thousand of samples) we need to estimate only

$$\frac{\left| d(C_R, C_S) - d\left(\widetilde{C}_R, \widetilde{C}_S\right) \right|}{d(C_R, C_S)} \tag{2}$$

Due to the metric properties

$$d\left(\widetilde{C}_R, \widetilde{C}_S\right) \leq d\left(\widetilde{C}_R, C_R\right) + d\left(C_R, C_S\right) + d\left(C_S, \widetilde{C}_S\right) \tag{3}$$

Then

$$\frac{d\left(\widetilde{C}_R, \widetilde{C}_S\right) - d(C_R, C_S)}{d(C_R, C_S)} \leq \frac{d\left(\widetilde{C}_R, C_R\right) + d\left(\widetilde{C}_S, C_S\right)}{d(C_R, C_S)} \tag{4}$$

Due to the symmetry the same inequality can be obtained with the opposite sign and as the result we get

$$\frac{\left| d\left(\widetilde{C}_R, \widetilde{C}_S\right) - d(C_R, C_S) \right|}{d(C_R, C_S)} \leq \frac{d\left(\widetilde{C}_R, C_R\right) + d\left(\widetilde{C}_S, C_S\right)}{d(C_R, C_S)}, \tag{5}$$

where the numerator corresponds to the accuracy of the estimation of covariance matrices and the denominator corresponds to the distance between covariance matrices for real and synthetic data. Thus, larger values of $d(C_R, C_S)$ result in a smaller relative error of the FID estimation. Experimentally, we compute the distances $d(C_R, C_S) = \text{tr}(C_R + C_S - 2\sqrt{C_R C_S})$ based on SwAV and InceptionV3 embeddings for StyleGAN2 trained on the Church dataset.[3] We obtain 0.083 for SwAV and 0.028 for InceptionV3, which confirms the validity of calculations above and explains the better sample-efficiency of SwAV presented in Figure 4.

## F    HUMAN EVALUATION

As an additional evidence that inter-image distances induced by SwAV are better aligned with human perception compared to InceptionV3, we perform two crowdsourcing experiments. All the data and labellings are released on the GitHub.[4].

**SwAV-based Precision and Recall have higher agreement with human judgements.** The key step of computing Recall is checking, if for a given real embedding $r$ there exists a generated embedding $g$ that is closer to $r$ than its $k$-th real neighbor. To verify, if a particular embedding agrees well with human perception, we perform the following procedure. We form a triplet of an anchor

---

[3]Since the absolute values of $d(C_R, C_S)$ depend on the scale of SwAV/InceptionV3 activations, we normalize them by the geometric mean of norms $C_R$ and $C_S$.

[4]https://github.com/stanis-morozov/self-supervised-gan-eval

| | CELEBAHQ, StyleGAN* | | CELEBAHQ, PGGAN | |
| --- | --- | --- | --- | --- |
| | Precision | Recall | Precision | Recall |
| InceptionV3 | 0.396 | 0.497 | 0.399 | **0.482** |
| SwAV | **0.418** | **0.525** | **0.482** | 0.479 |

| | LSUN-Bedroom, StyleGAN* | | LSUN-Bedroom, PGGAN | |
| --- | --- | --- | --- | --- |
| | Precision | Recall | Precision | Recall |
| InceptionV3 | 0.417 | 0.522 | 0.375 | **0.444** |
| SwAV | **0.450** | **0.530** | **0.446** | 0.429 |

| | LSUN-Church, MSG | | LSUN-Church, PGGAN | |
| --- | --- | --- | --- | --- |
| | Precision | Recall | Precision | Recall |
| InceptionV3 | 0.386 | 0.473 | 0.400 | 0.449 |
| SwAV | **0.477** | **0.497** | **0.445** | **0.458** |

Table 7: Human evaluation of the precision and recall agreement with different embeddings (higher is better).

real image $I_{\text{anchor}}$ that contributes to the Recall value, its 5-th closest neighbor among the real images $I_{\text{5th}}$ and the generated image $I_{\text{gen}}$ that appears to be closer to $I_{\text{anchor}}$ in terms of the considered embedding. A human assessor is then asked to choose an image between $I_{\text{gen}}$ and $I_{\text{5th}}$ that is more similar to $I_{\text{anchor}}$. Once the assessor chooses the generated one, we consider it as a case of agreement with the embedding. The embeddings with higher agreement rate are more suitable for computing Recall.

For Precision, we similarly form the triplets consisting of a generated image, its 5-th neighbor among the generated images and a real image $I_{\text{real}}$ closer to it $I_{\text{anchor}}, I_{\text{5th}}, I_{\text{real}}$. Once an assessor answers that $I_{\text{real}}$ is more similar to $I_{\text{anchor}}$ then $I_{\text{5th}}$, we consider this is as an agreement with the embedding.

Here we always use the same real and generated samples as for the evaluation of the metrics in Section 3. We label three datasets with two GAN models for each. For each pair of a dataset and a generator, we label 200 different triplets, each by ten different assessors. An assessor is also able to choose the options "equally similar" or "both completely dissimilar". Once the "equally similar" is chosen, we suppose that the agreement happens with the probability $0.5$. The user interface is illustrated on Figure 8 (left). All the labeling was performed in Yandex Toloka [5]. The results are presented on Table 7 and confirm that SwAV emebddings mostly have higher agreement with human perception.

**Quality of neighbors.** As a more simple experiment, we also ask human assessors to compare quality of top-5 neighbors produced by InceptionV3 and SwAV embeddings. Namely, we take a set of $N$ real images $\mathcal{I}$, same as in Section 3. For a given real image $r \in \mathcal{I}$ we form two lists of its 5 nearest neighbors $\mathcal{B}_{\text{IV3}} \subset \mathcal{I}$ and $\mathcal{B}_{\text{SwAV}} \subset \mathcal{I}$ based on InceptionV3 and SwAV embeddings. An assessor is asked to assign $r$ either to $\mathcal{B}_{\text{IV3}}$ or to $\mathcal{B}_{\text{SwAV}}$. Same as above, the assessor may also label it as "equal" which is treated as an equal probability of each set to be chosen. The user interface is illustrated on Figure 8 (right). For each dataset we form 500 different triplets $r, \mathcal{B}_{\text{SwAV}}, \mathcal{B}_{\text{IV3}}$, each labeled by ten different assessors. Table 8 presents probabilities that assessors prefer a SwAV-based set of neighbors, indicating that SwAV-induced distances better capture perceptual similarity compared to InceptionV3.

| | InceptionV3 | SwAV |
| --- | --- | --- |
| CELEBAHQ | 0.37 | **0.63** |
| LSUN-Bedroom | 0.39 | **0.61** |
| LSUN-Church | 0.41 | **0.59** |

Table 8: Comparison of top-5 neighbor lists quality based on SwAV/InceptionV3 representations.

---

[5]https://toloka.ai

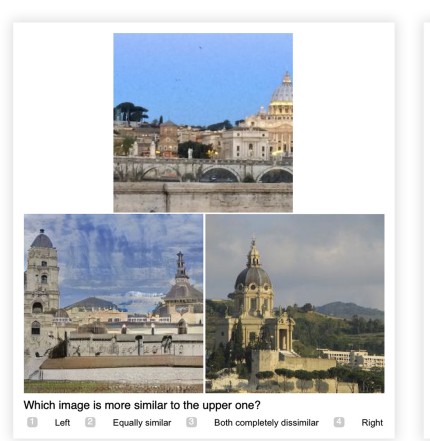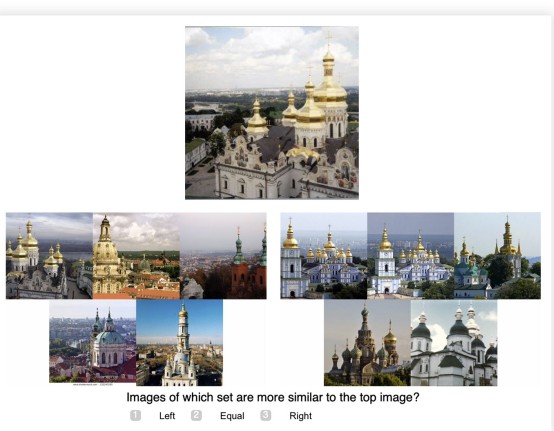

Figure 8: User interface for precision / recall agreement labeling (left) and top-5 neighbours labeling (right).

| model | source | truncation |
|---|---|---|
| CELEBAHQ ||| 
| StyleGAN | https://github.com/NVlabs/stylegan | - |
| StyleGAN* | https://github.com/NVlabs/stylegan | 0.7 |
| MSG | https://github.com/akanimax/msg-stylegan-tf | - |
| MSG* | https://github.com/akanimax/msg-stylegan-tf | 0.6 |
| PGGAN | https://github.com/tkarras/progressive_growing_of_gans | - |
| FFHQ ||| 
| StyleGAN2 | https://github.com/NVlabs/stylegan2 | - |
| StyleGAN | https://github.com/rosinality/style-based-gan-pytorch | - |
| MSG | https://github.com/akanimax/msg-stylegan-tf | - |
| MSG* | https://github.com/akanimax/msg-stylegan-tf | 0.6 |
| LSUN-Bedroom ||| 
| StyleGAN | https://github.com/NVlabs/stylegan | - |
| StyleGAN* | https://github.com/NVlabs/stylegan | 0.7 |
| PGGAN | https://github.com/tkarras/progressive_growing_of_gans | - |
| RPGAN | https://github.com/anvoynov/RandomPathGAN, model: generator_lsun_2 | - |
| RPGAN-div | https://github.com/anvoynov/RandomPathGAN, model: generator_lsun_high_diversity | - |
| COCO-GAN | https://github.com/hubert0527/COCO-GAN/tree/12b90e26e23214c2072c9701644e9724e052743c, model: LSUN_256x256_N2M2S128 | - |
| LSUN-Church ||| 
| StyleGAN2 | https://github.com/NVlabs/stylegan2 | - |
| StyleGAN2* | https://github.com/NVlabs/stylegan2 | 0.5 |
| PGGAN | https://github.com/tkarras/progressive_growing_of_gans | - |
| MSG | https://github.com/akanimax/msg-stylegan-tf | - |
| MSG* | https://github.com/akanimax/msg-stylegan-tf | 0.6 |
| SNGAN | Submission authors implementation | - |

Table 9: The URLs with GAN checkpoints and truncation levels used in our experiments.

| Representation | Dim | URL | Checkpoint |
|---|---|---|---|
| InceptionV3 | 2048 | `https://github.com/mseitzer/pytorch-fid` | `https://github.com/mseitzer/pytorch-fid/releases/download/fid_weights/pt_inception-2015-12-05-6726825d.pth` |
| Resnet-50 | 2048 | `https://pytorch.org/docs/stable/torchvision/models.html` | `https://download.pytorch.org/models/resnet50-19c8e357.pth` |
| Imagenet21k | 2048 | `https://tfhub.dev/google/bit/m-r50x1/1` | `https://tfhub.dev/google/bit/m-r50x1/1` |
| SwAV | 2048 | `https://github.com/facebookresearch/swav` | `https://dl.fbaipublicfiles.com/deepcluster/swav_800ep_pretrain.pth.tar` |
| DeepClusterV2 | 2048 | `https://github.com/facebookresearch/swav` | `https://dl.fbaipublicfiles.com/deepcluster/deepclusterv2_800ep_pretrain.pth.tar` |
| MoCoV2 | 2048 | `https://github.com/facebookresearch/moco` | `https://dl.fbaipublicfiles.com/moco/moco_checkpoints/moco_v2_800ep/moco_v2_800ep_pretrain.pth.tar` |

Table 10: The URLs with checkpoints used in our experiments.

