# OpenReview forum: "On Self-Supervised Image Representations for GAN Evaluation"
_ICLR.cc/2021/Conference — ICLR 2021 Spotlight_

### Official Review · AnonReviewer3 · 2020-10-25

**Rating:** 7
**Confidence:** 4

**Review:**

Overview of paper: this work compares supervised feature extractors vs. two types of self-supervised feature extractors for the task of GAN model evaluation. It shows that the ranking provided by self-supervised features is different from that of supervised features, and claims it corresponds better with human judgement. Experiments are conducted of multiple large GANs and datasets.

Novelty: I am not aware of previous works investigating self-supervised features for GAN evaluation, but note that [1] evaluated self-supervised features as a perceptual loss which is highly related.

Significance: the current metrics used to evaluate GANs are well-known to be problematic and the search for better measures is important. On the other hand, I am not certain that this paper is conclusive enough be able to shift the community towards different metrics. This is a hard thing to do as even changing the evaluation from using InceptionV3 features (which are fairly outdated) to more modern ResNets has not happened yet.

Methodology: I have a few issues with the method - although the authors claim to have better agreement with human and objective judgements, this is not extremely well justified. E.g. showing that Swav can classify facial attributes between InceptionV3 is not by itself very indicative - it uses a better architecture and more generally transfer learning of self-supervise vs. supervised methods was extensively validated in the original papers (and is about equal). Also, supervised ImageNet features are particularly poor for faces, I guess that for other objects types results might be different. The evidence for the "groundtruth" ranking for precision and recall is not particularly strong.

Evaluation:  + many gans and datasets evaluated - why only SWAV and DeepClusters, why not use the other popular contrastive learning methods e.g. MoCo, SimCLR, BYOL?

Overall: the investigation of better ways of evaluating GANs is important. The main criticism is that (in my opinion) not enough effort was taken to establish the groundtruth ranking between models, making the results of this investigation less significant.

########################################################################################

The rebuttal addressed my concerns - I increased the score

[1] Zhang et al. ,The unreasonable effectiveness of deep features as a perceptual metric, CVPR'18

---

> ### Author Response · Authors · 2020-11-25
> **Rebuttal**
>
> We thank the reviewer for the review and constructive comments. Here we address each of the concerns:
>
> [shifting the community towards different metrics is a hard thing to do as even changing the evaluation from using InceptionV3 features (which are fairly outdated) to more modern ResNets has not happened yet.] In fact, our experiments show that shifting from InceptionV3 to classification-trained Resnet50 is not needed since they provide the same GAN ranking. The actual difference appears between InceptionV3 and the state-of-the-art self-supervised models, as many cases of misleading comparison are corrected for several datasets. Furthermore, in Section 4.3 we demonstrate that SwAV-based evaluation is more sample-efficient compared to InceptionV3,  which is a very appealing property, as few-shot generation becomes an active research topic.
>
> [why not use the other popular contrastive learning methods?] We chose SwAV/DeepCluster since they provided the best transfer accuracy at the moment of submission. In the new revision, we also added the MoCOv2 model, which features are inferior to Swav in terms of transfer. Compared to InceptionV3, MoCOv2 corrects several misleading FID ranking cases on LSUN-Bedroom while providing the same ranking on CelebA (which is corrected by the more powerful Swav model).
>
> ["groundtruth" ranking] We agree with this point. As additional evidence, we perform a human evaluation to obtain the groundtruth for the Precision metric, which quantifies the visual quality. Specifically, for StyleGAN2 and MSG trained on LSUN-Church, we ask ten evaluators to predict if a given image is real or fake (each evaluator marked 500 images, and we use the evaluation setup as in [A]). We chose this model pair since InceptionV3 ranks them as comparable, while SwAV is confident that StyleGAN2 samples are more realistic. The average prediction error rate is 2.8% for StyleGAN2 and 0.4% for MSG, indicating that the groundtruth precision of StyleGAN2 is higher, which is consistent with the SwAV ranking.
>
> [A] Improved techniques for training gans, NIPS 2016

---

### Official Review · AnonReviewer1 · 2020-10-27
**Interesting empirical evaluation**

**Rating:** 7
**Confidence:** 4

**Review:**

This paper provides an interesting empirical study of self-supervised image embeddings as an alternative to pre-trained ImageNet classification models, for the purpose of evaluating the quality and diversity of GAN generative image models. I always found it a little odd that a model trained on ImageNet with cross-entropy loss should somehow be a magical universal quality metric, and I am happy to see that this paper provides good evidence that this is not the case. The authors select 2 self-supervised models and compare them against a number of supervised models. The metrics used are FID and Precision/Recall. I am curious why Inception Score was not also compared?

The paper does quite a thorough job of selecting and comparing models, by normalizing for architecture and changing dataset or loss function. It shows clearly that self-supervised methods outperform the supervised methods for ranking various GAN models.

It would have been interesting to train the self-supervised model on the dataset itself e.g. LSUN or CelebA to see whether that provides an even more useful signal. Given that deep networks find it hard to generalize across datasets, I would expect that directly training an embedding on the target dataset would do better. Did the authors try something along these lines?

A minor comment is that the layout of the results and comments is a bit confusing: due to the very long number of points that refer to a particular figure and needing lots of scrolling back and forth. Some better way to organize the information and comments would be appreciated.

I would also find it insightful to better understand *why* self-supervision works better for evaluating representations? Any comments to this regard would be interesting. Lastly, I am curious why the authors did not consider self-supervised methods such as SimCLR?

I have read the rebuttals and other comments and maintain my rating of the paper.

---

> ### Author Response · Authors · 2020-11-25
> **Rebuttal**
>
> We thank the reviewer for the review and address the questions below:
>
> [I am curious why Inception Score was not also compared?] In our submission, we mostly focus on the non-Imagenet domains, where IS was shown to be inadequate [A]. Furthermore, FID/Precision/Recall are currently used more broadly in the GAN literature.
>
> [A] A Note on the Inception Score, ICML 2018 Workshop on Theoretical Foundations and Applications of Deep Generative Models
>
> [train the self-supervised model on the dataset itself. Did the authors try something along these lines?] Using a separate embedding model for each dataset definitely can result in a more accurate evaluation. However, this would make the research process more complicated since for each new dataset, an embedding model should be trained and maintained for further consistent usage in the community. In contrast, the proposed Swav/DeepCluster embeddings are designed to provide high transfer quality, therefore, can be used universally.
>
> [better understand why self-supervision works better for evaluating representations?] The main idea is that classification-pretrained embeddings (e.g., InceptionV3) capture only the most discriminative image features needed to predict the class labels. In contrast, self-supervised embeddings only learn to be invariant to simple augmentations, preserving more image-specific information, often required to discriminate between real/fake. The results from Figure 2 and Table 2 support this intuition experimentally.
>
> [why the authors did not consider self-supervised methods such as SimCLR?] We chose SwAV/DeepCluster since they provided the state-of-the-art transfer accuracy at the moment of submission. In the new revision, we also added the MoCOv2 model, which features are inferior to Swav in terms of transfer. Compared to InceptionV3, MoCOv2 corrects several misleading FID ranking cases on LSUN-Bedroom while providing the same ranking on CelebA (which is corrected by the more powerful Swav model). Overall, we observe that more transferable features are better for GAN evaluation as well.

---

### Official Review · AnonReviewer4 · 2020-10-27

**Rating:** 7
**Confidence:** 4

**Review:**

## Summary

The papers looks at the problem of evaluating GAN samples. Current methods, such as FID/PR with Inception v3 are problematic because they generally depend on using the features of a model discriminatively trained on (a super set of) ImageNet. The authors show that these type of models ignore details that are meaningful when for example comparing results on CelebaHQ.

Instead the authors propose to use a recent self-supervised trained model which have been shown to provide more general representations. They take a selection of recent powerful GANs, and compare the ranking of their results based on FID/PR with discriminative imagine features vs self-supervised features and show that there are indeed differences.

To evaluate the ground truth, the authors device a number of small experiments that attempt to establish these facts: retrieving celebA labels from features of each model and an additional classifier trained on one GAN output evaluated on another. In all experiments the authors show that the self-supervised trained model produces a GAN ranking that is closer to the truth.

## Review

The paper is well written and provides a very nice overview of recent advances in GANs and description of their evaluation methods. While the proposed method is a simple improvement over previous work (replace the feature extractor, keeping most else constant), the empirical evaluation is very thorough and well done.

In particular I found the additional experiments based on the results in table 1 and 3 very informative and welcome. Results in table 2 and 4 give a very interesting confirmation that self-supervised embeddings are indeed more informative.

The visualisations are well done and relevant, and the markings in the table make finding comparisons straightforward.

## Discussion

While the ordering between SwAV and DeepClusterV2 is the same, the actual numbers are significantly different, do the authors know why this might happen?

The authors compare to two specific self-supervised algorithms, which are additionally trained using clustering, is there a particular reason those models were chosen? Does the type of contrastive learning impact the results, or is generally better representations (as measured by fine-tuning for imagenet) better for GAN evaluation too? Would it be worthwhile to attempt to add the types of artifacts in GANs to the set of augmentations done for the contrastive learning?

In general, I think the paper can benefit from more analysis around the choice for the right self-supervised network and trade offs.

## Post answer

My questions are appropriately answered and I appreciate the addition of section 3.4. I think my current score accurately reflects my evaluation of the paper, with the remaining concern being the magnitude of the contribution.

---

> ### Author Response · Authors · 2020-11-25
> **Rebuttal**
>
> We thank the reviewer for the review and address each of the questions below:
>
> [Actual numbers for SwAV/DeepCluster are significantly different] The reason is that the FID metric values are not invariant to the typical scale of embedding activations, which can be different in different models. Note that this issue does not arise for Precision and Recall, and the numbers are very close.
>
> [is there a particular reason Swav/DeepCluster were chosen?] We chose these models since they provided state-of-the-art transfer accuracy at the moment of submission. To extend the number of models, we also added the MoCOv2 model in the new revision, which features are inferior to Swav in terms of transfer performance. Compared to InceptionV3, MoCOv2 corrects several misleading FID ranking cases on LSUN-Bedroom while providing the same ranking on CelebA (which is corrected by the more powerful Swav model). Overall, we observed that the more transferable features are better for GAN evaluation as well.
>
> [Would it be worthwhile to attempt to add the types of artifacts in GANs to the set of augmentations done for the contrastive learning?] This is an interesting suggestion. We suppose that such augmentations can be used to produce the more appropriate negatives for contrastive approaches. However, it is not clear if the artifacts from different GAN models are the same, therefore, there is a risk of “overfitting” to a particular set of artifacts.

---

### Official Review · AnonReviewer2 · 2020-10-28
**Interesting proposal with some missing experiments**

**Rating:** 7
**Confidence:** 4

**Review:**

This paper proposes to use image representations from trained self-supervised models to evaluate GANs more accurately. Compared to the currently used representations from supervised-pretrained models e.g. InceptionV3, the authors claim, that such embeddings suppress information not critical for the classification process which, however, are assumed to be crucial for assessing the full distributions of real and generated images. The authors use 5 datasets and their respective representations from 5 models, 3 supervised and 2 self-supervised, to show that representations from self-supervised models lead to better GAN evaluations. The representations were used to evaluate 6 GAN models with 3 metrics, namely FID, Precision and Recall. A ranking of the GAN models shows inconsistencies between supervised and self-supervised based representations. By visual inspection, prediction accuracy tests, and a comparison of representation invariances the authors show that rankings via self-supervised embeddings are more plausible.

Pros: Interesting proposal to better evaluate GANs and generative models for image data. The paper is well written and easy to understand. The experiments are extensive and support the claim of the authors. Testing for invariances of representations is an interesting idea and the results support the use of embeddings from self-supervised models.

Cons: The authors argue that latent representations from an autoencoder capture all the information from images. It would be interesting to see how such representations, e.g. from the autoencoder used to show the invariances described in section A.1, behave compared to the proposed self-supervised representations. I would like to see them to be included in the experiments.

Minor comment: Typo in A.1: corrseponding

Edit: The authors have not responded to any of the reviews, i lower my rating to 4

Edit2: Oh there was a misunderstanding, i probably was not logged in and didn't see any comments and reviews. I raised the rating and will read the answers and will rate again.

Edit3: After reading the rebuttals, i raise my rating to 7

---

> ### Author Response · Authors · 2020-11-25
> **Rebuttal**
>
> We are somewhat rudely surprised that the reviewer has lowered the score before the end of the Discussion phase. The reviewer requested the additional experiments, and it requires time to perform them scrupulously. But we definitely aim to provide a thorough response as reflected in our answers and the new revision.
>
> The only weakness mentioned by the reviewer is that it would be interesting to include the autoencoder representations into comparison.
>
> While autoencoders can be used for analysis, we argue that their representations are not well-suited to evaluate GANs. We confirm this claim experimentally in Appendix D of the new revision.
>
> The main idea is that the autoencoder’s training objective includes L2 reconstruction terms, therefore, their embeddings place too much emphasis on the exact spatial arrangement.
> Meanwhile, GANs are evaluated on finite datasets, which can lack images of the same spatial structure. We highlight this issue in Figure 7 in Appendix D. For several real images, we demonstrate their nearest neighbors in terms of SwAV and autoencoder’s embeddings. In the latter case, the neighbors are often clearly unrealistic samples with the same layout, resulting in misleading evaluation, for instance, via Precision and Recall metrics. To sum up, the distance in the space of the autoencoder’s embeddings is less informative for analysis of the real/fake distributions.

---

### Public Comment · ~Xun_Huang1 · 2020-11-10
**Some empirical observations when using SwAV metrics in my experiments.**

As a researcher working on GANs, I would like to share some of my observations when using self-supervised representations (SwAV in particular) to evaluate models during my experiments, after reading this paper.

1. First, I generally agree that SwAV is better than Inception-v3 as a backbone network for GAN evaluation and am convinced to shift towards using it.
2. SwAV is significantly better than Inception-v3 at distinguishing real and fake samples. The 1-NN accuracy [1] with SwAV is much higher than that with Inception-v3, (e.g., 0.95 vs 0.7). I hypothesize that this would hold for a linear classifier as well, although I haven't tried it. The gap between 1-NN fake accuracy is even larger (e.g., 0.999 vs 0.75), which suggests partial mode collapse is detected by SwAV but not by Inception-v3. The paper briefly mentions similar results when comparing precision/recall, but I believe this phenomenon can be investigated further. Is it possible to conduct simulated experiments to show that SwAV is more sensitive to mode collapse than Inception-v3? For example, one way to simulate a mode collapsed version of an image set is to randomly pick 100 samples from 10000, and generate 100 images from each sample using small data augmentation.
3. During training, I find the curve of SwAV is a bit smoother than that of Inception-v3. I think it's a good property because we generally expect the generator to be improving throughout the training. I am wondering if the authors also observe this and can quantify it.

[1] Xu, Qiantong, et al. "An empirical study on evaluation metrics of generative adversarial networks." arXiv preprint arXiv:1806.07755 (2018).

---

> ### Author Response · Authors · 2020-11-25
> **Rebuttal**
>
> Thanks for you interest to our work. We attribute the curve smoothness to the fact that SwAV-based FID computation is much more stable with respect to the sampling stochasticity. We elaborate on this effect in section 4.3 and Appendix E of the revised submission.

---

### Author Response · Authors · 2020-11-25
**Rebuttal**

We thank the reviewers for their time and useful suggestions. We have uploaded a new revision of our paper with several changes described in the individual answers below. Note that we have added a new section 4.3 and Appendix E, which demonstrate the exceptional sample-efficiency of SwAV-based FID computation.

---

### Decision · Program_Chairs · 2021-01-07
**Final Decision**

**Decision:**

Accept (Spotlight)

**Comment:**

All four reviewers unanimously recommended for an acceptance (four 7s). They generally appreciated that the proposed idea is novel and experiments are convincing. I think the paper tackles an important problem of evaluating GANs, and the idea of using self-supervised representations, as opposed to the conventional ImageNet-based representations, would lead to interesting discussions and follow-ups.